# Antiphospholipid antibodies and vitamin D deficiency in COVID-19 infection with and without venous or arterial thrombosis: A pilot case-control study

Ruchi Shah[1][○], Yaqub Nadeem Mohammed[1][‡], Tracy J. Koehler[2][○], Jasmeet Kaur[1][‡], Margarita Toufeili[1][‡], Priyanjali Pulipati[1][‡], Ahmed Alqaysi[1][‡], Ali Khan[1][‡], Mahrukh Khalid[1][‡], Yi Lee[1][‡], Parveen Dhillon[3][‡], Anna Thao Dan[3][‡], Nicholas Kumar[3][‡], Monica Bowen[2][‡], Anupam A. Sule[1][○], Geetha Krishnamoorthy[1][○]*

1 Department of Internal Medicine, Saint Joseph Mercy Oakland, Pontiac, Michigan, United States of America, 2 Mercy Health Muskegon, Muskegon, Michigan, United States of America, 3 Ross University School of Medicine, Miramar, Florida, United States of America

○ These authors contributed equally to this work.
‡ YNM, JK, MT, PP, AA, AK, MK, YL, PD, NK, ATD, and MB also contributed equally to this work.
* geetha.krishnamoorthy@stjoeshealth.org

**Data Availability Statement:** All relevant data are within the paper and its Supporting Information files.

## Abstract

### Background

Coronavirus disease-2019 (COVID-19) is associated with thromboembolism. Antiphospholipid antibody (APLa) formation is one of the mechanisms. Vitamin D deficiency has been associated with thrombosis in antiphospholipid antibody syndrome.

### Objective

Measure APLa and vitamin D in hospitalized COVID-19 patients with and without thrombosis to evaluate if thromboembolism is associated with concomitant APLa and vitamin D deficiency.

### Methods

Case-control study. Hospitalized COVID-19 patients with a thromboembolic event (ischemic stroke, myocardial infarction, deep venous thrombosis/pulmonary embolism, Cases n = 20). Controls (n = 20): Age, sex-matched without thromboembolic events. Patients with autoimmune disorders, antiphospholipid antibody syndrome, thrombophilia, anticoagulation therapy, prior thromboembolism, chronic kidney disease 3b, 4, end-stage renal disease, and malignancy were excluded. Given the limited current literature on the role of concomitant antiphospholipid antibodies and vitamin D deficiency in causing venous and/or arterial thrombosis in hospitalized COVID-19 patients, we enrolled 20 patients in each arm. Anti-cardiolipin IgG/IgM, beta-2 glycoprotein-1 IgG/IgM, lupus anticoagulant and vitamin D levels were measured in both groups.

**Funding:** The author(s) received no specific funding for this work.

**Competing interests:** The authors have declared that no competing interests exist.

## Results

Cases were 5.7 times more likely to be vitamin D deficient (OR:5.7, 95% CI:1.3–25.6) and 7.4 times more likely to have any one APLa (OR:7.4, 95% CI: 1.6–49.5) while accounting for the effects of sex. Patients with both APLa and vitamin D deficiency had significantly more thrombosis compared to patients who were antibody positive without vitamin D deficiency (100% vs 47.4%; p = 0.01).

## Conclusions

Thrombosis in COVID-19 was associated with concomitant APLa and vitamin D deficiency. Future studies in COVID-19 should assess the role of vitamin D in reducing thrombosis.

## Introduction

Coronavirus Disease 2019 (COVID-19) due to severe acute respiratory syndrome coronavirus 2 (SARS-Co-V 2) is associated with a prothrombotic state, leading to both venous thromboembolism (VTE) as well as arterial thromboses such as stroke and myocardial infarction [1, 2].

Zhang et al., first described a case series of three patients with COVID-19 with limb ischemia, who tested positive for anticardiolipin IgA and IgG and anti-beta-2 glycoprotein 1 IgA and IgG [3]. Recently, more researchers found that in both antiphospholipid antibody syndrome and COVID-19 infection, systemic inflammatory response causes the release of cytokines, and complement activation leading to endothelial damage and clot formation [4–6]. It is postulated that antiphospholipid antibodies (APLa) may be one of the mechanisms of thrombosis in patients with COVID-19 infection [7]. A recent study tested eight types of APLa, namely anticardiolipin Immunoglobulin (Ig)A/IgG/IgM, anti-beta-2 glycoprotein 1 IgA/IgG/IgM, and anti-phosphotidylserine/prothrombin IgM/IgG in the sera of 172 patients with COVID-19 [8]. In this study, the authors found that 52% of patients with COVID-19 had at least one APLa [8]. Bowles et al., reported that among hospitalized COVID-19 patients with prolonged activated partial thromboplastin time (aPTT), 91% were positive for lupus anticoagulant [7]. A meta-analysis that included 21 studies (total n = 1159) reported that the most common APLa is lupus anticoagulant (50.7%); furthermore, 28.8% of critically ill patients had anticardiolipin (IgG or IgM) and 12% of them had anti-beta-2 glycoprotein 1 (IgG or IgM) [9].

T regulatory lymphocytes (Tregs) are low in severe viral respiratory diseases, including COVID-19 [10]. Tregs level and function increase with vitamin D supplementation [11]. Higher vitamin D level leads to elevated Treg/total T lymphocyte ratio resulting in a more immune-suppressive phenotype [11]. In addition to this immunomodulatory effect, vitamin D deficiency may play a role in the development of VTE [12]. In an animal study by Aihara et al., vitamin D receptors were shown to have a physiological role in the antithrombotic hemostasis [13]. A large prospective Danish study that included 18791 patients with a 30-year follow-up showed that reduced 25-hydroxy vitamin D concentrations are associated with an increased risk of venous thromboembolism [14].

A study by Agmon-Levin et al., measured vitamin D levels in 179 patients with antiphospholipid antibody syndrome and 141 healthy controls and found that vitamin D deficiency was seen in 49.5% of patients with antiphospholipid antibody syndrome compared to 30% of controls (p < 0.001) [15]. In addition, the authors found vitamin D deficiency correlated with thrombosis (58% vs 42%; p < 0.05) in patients with antiphospholipid antibody syndrome and showed that vitamin D inhibited anti-beta 2-glycoprotein antibody induced Tissue factor (TF) expression in vitro [15].

Since APLa have been demonstrated in COVID-19 infection, and vitamin D deficiency has been shown to correlate with thrombosis in patients with antiphospholipid antibody syndrome, we decided to measure APLa and vitamin D levels in hospitalized patients with COVID-19 with and without venous or arterial thromboses to evaluate if the concomitant presence of APLa and vitamin D deficiency was associated with venous and arterial thromboembolic events.

## Methods

This pilot case-control study was reviewed and approved by the Institutional Review Board of Saint Joseph Mercy Oakland Hospital, Pontiac, Michigan. Written informed consent was obtained from all the study participants in accordance with the consent procedure approved by the Institutional Review Board.

### Setting

Patients hospitalized to the medical floor or intensive care unit at Saint Joseph Mercy Oakland Hospital, a 497-bed community teaching hospital in Pontiac, Michigan between January 2021 and May 2021.

### Participants

Patients aged ≥18 years hospitalized for COVID-19 confirmed by real time reverse transcription polymerase chain reaction (RT-PCR) between January 2021 to May 2021 with a thromboembolic event including, ischemic stroke diagnosed by Neurologist, and Magnetic Resonance Imaging (MRI) or Computerized Tomography (CT), acute coronary syndrome—non ST-elevation myocardial infarction (NSTEMI) or ST-elevation myocardial infarction (STEMI) diagnosed by a cardiologist with intracoronary thrombosis found during cardiac catheterization, limb ischemia or deep venous thrombosis (DVT) diagnosed by arterial or venous doppler ultrasound, or pulmonary embolism (PE) diagnosed by ventilation/perfusion scan or CT angiography of Chest were identified and enrolled in the study (Case group). Controls were age and sex-matched to patients ≥18 years hospitalized for COVID-19 confirmed by RT-PCR during the same timeframe who did not experience a thromboembolic event. Informed consent was obtained prior to study enrollment.

Exclusion criteria were the following: History of systemic lupus erythematosus, antiphospholipid antibody syndrome, rheumatoid arthritis, or any one of the inherited hypercoagulable states, patients on chronic anticoagulation with low molecular weight heparin, warfarin, or direct acting oral anticoagulants, prior history of DVT or PE in patients with venous thrombosis, prior stroke seen in CT or MRI, prior NSTEMI or STEMI in patients with stroke or acute coronary syndromes, severe peripheral arterial disease in patients with limb ischemia, chronic kidney disease stages (CKD) 3b, 4 and end stage renal disease (ESRD) on renal replacement therapy, and any active malignancy. Ours is an exploratory study for hypothesis generation that concomitant presence of vitamin D deficiency and APLa in hospitalized patients with COVID-19 infection may increase the risk for venous and/or arterial thrombosis. Given the limited current literature on the role of concomitant APLa and vitamin D deficiency in hospitalized COVID-19 patients, we enrolled 20 patients in each arm (n = 40 in total).

### Study variables

Data collection from electronic medical records included age, sex, race, body mass index, comorbidities including hypertension, coronary artery disease, diabetes mellitus, peripheral

arterial disease, atrial fibrillation, CKD (Stage 1-3a), type of thromboembolic event and the type/results of diagnostic test and management in case group, complete blood count with differential count, prothrombin time (PT), aPTT, liver enzymes and inflammatory markers (Ferritin, C-Reactive Protein (CRP), D-dimer).We made efforts to eliminate sources of bias by matching our cases and controls for age and sex, and comorbidities as indicated in Table 1.

## Testing

Venipuncture was performed for testing anti-cardiolipin IgG/IgM, beta-2 glycoprotein-1 IgG/IgM, lupus anticoagulant by Dilute Russel Viper Venom test and serum 25-hydroxy vitamin D level. The actual tests were done using commercially available laboratory tests. If patients were vitamin D deficient and/or antiphospholipid antibodies were identified, the attending physician was notified.

**Primary measures.** The groups were assessed for differences regarding frequency of APL antibody positivity, defined as a positive screening for any of the following: Anti-cardiolipin IgM antibody (positive >12.5 MPL), anti-cardiolipin IgG antibody (positive >15 GPL), beta-2 glycoprotein 1 IgM antibody (positive >20 SMU), beta-2 glycoprotein 1 IgG antibody (positive >20 SGU), Dilute Russell Viper Venom Test to identify lupus anticoagulant. Frequency of vitamin D deficiency was defined as 25 hydroxy vitamin D level < 20 ng/ml (50 nmol/L).

**Statistical analysis.** Comparisons between cases and controls were made using the t-test (normally distributed variables) or Mann Whitney (non-normally distributed variables) for quantitative data and are shown as the mean+Standard Deviation (SD) or median [interquartile range]. Nominal variables were assessed using the chi-square or Fisher's exact test and are reported as percentages. The Cochran-Mantel-Haenszel test was used to control for the effects of sex when deriving the Odds Ratios (OR) and 95% Confidence intervals (CI) for the primary measures. Significance was assessed at $p < 0.05$. Data were analyzed using IBM SPSS Statistics v 23 (Armonk, NY: IBM Corp).

**Table 1. Demographic and clinical characteristics.**

|  | Case n = 20 | Control n = 20 | p-value |
|---|---|---|---|
| Age* | 59.7+16.3 | 59.8+15.7 | 0.98 |
| Sex* |  |  | >0.999 |
| Female | 40% (8) | 40% (8) |  |
| Male | 60% (12) | 60% (12) |  |
| BMI | 33.6+8.3 | 31.8+6.3 | 0.45 |
| Race/Ethnicity |  |  | >0.999 |
| Black | 25% (5) | 30% (6) |  |
| White | 70% (14) | 70% (14) |  |
| Unknown | 5% (1) | 0% (0) |  |
| Hypertension, % Yes | 60% (12) | 50% (10) | 0.53 |
| Diabetes, % Yes | 25% (5) | 20% (4) | >0.999 |
| Hyperlipidemia, % Yes | 55% (11) | 40% (8) | 0.34 |
| PAD, % Yes | 10% (2) | 0% (0) | 0.49 |
| CHF, % Yes | 5% (1) | 0% (0) | >0.999 |
| CKD, % Yes | 15% (3) | 0% (0) | 0.23 |
| Atrial fibrillation, % Yes | 5% (1) | 5% (1) | >0.999 |

*Matching variables; BMI: Body Mass Index, PAD: Peripheral arterial disease, CHF: Congestive heart failure, CKD: Chronic Kidney Disease.

## Results

Table 1 shows the results of demographic and clinical characteristic comparisons between cases and controls. The groups were well matched on age and sex and there were no statistically significant differences between the groups regarding other demographic or clinical characteristics.

Significant findings in admission laboratory evaluation included elevated neutrophil count, D-dimer, alkaline phosphatase, C- Reactive Protein (CRP), and lower hemoglobin in cases compared to controls (Table 2).

Of the 20 cases with arterial or venous thrombosis, 15 patients had PE, 3 patients had DVT, 1 patient had ischemic stroke and 1 patient had NSTEMI with intracoronary thrombosis found during cardiac catheterization.

### Antiphospholipid antibodies and vitamin D deficiency

Accounting for the effects of sex, cases were 5.7 times more likely to be Vitamin D deficient (OR: 5.7, [95% CI: 1.2–26.4]) and 7.4 times more likely to have any one positive APLa (OR: 7.4 [95% CI: 1.4–38.2]; Table 3) and 10.3 times more likely to have a positive finding for lupus anticoagulant (OR: 10.3 [95% CI: 1.3–79.5]; Table 3). No other findings were significant between the groups.

## Discussion

We found that APLa were present significantly more frequently in our hospitalized COVID-19 patients with thrombotic manifestations compared to patients without thrombotic manifestations. Vitamin D deficiency (level < 20 ng/mL) also was significantly more frequent in patients with thrombotic manifestations. All the patients with both APLa and vitamin D deficiency had thrombotic manifestations, whereas only half of the patients with APLa had thrombotic manifestation when they were not vitamin D deficient.

**Table 2. Admission Laboratory evaluation comparisons.**

| Admission labs | Case n = 20 | Control n = 20 | p-value |
|---|---|---|---|
| Hemoglobin (g/dL)* | 13.2+2.0 | 14.8+1.7 | 0.009¶ |
| eGFR (mL/min)* | 70.8+19.7 | 75.7+21.1 | 0.46 |
| Neutrophil count^ (10⁹ cells/L) | 9.5 [5.6–12.9] | 4.9 [3.0–6.7] | 0.002¶ |
| Lymphocyte count^ (10⁹ cells/L) | 0.8 [0.7–1.8] | 0.9 [0.7–1.2] | 0.64 |
| D-Dimer (ng/mL)^ | 1885 [651–5001] n = 19 | 314 [284–633] | <0.001¶ |
| Ferritin (mcg/L)^ | 421 [221–700] | 274 [124–1221] n = 19 | 0.81 |
| Aspartate amino transferase (U/L)^ | 33 [26–41] n = 19 | 44 [29–64] | 0.19 |
| Alanine amino transferase (U/L)^ | 46 [23–63] n = 19 | 32 [23–57] | 0.53 |
| Alkaline phosphatase (U/L)^ | 84 [64–106] n = 19 | 60 [48–67] | 0.003 |
| Total bilirubin (mg/dL)^ | 0.8 [0.7–1.2] | 0.7 [0.5–0.8] | 0.07 |
| Albumin (g/L)* | 2.8+0.9 n = 19 | 3.1+0.4 | 0.14 |
| Partial thromboplastic time (sec)^ | 31 [27–62] n = 18 | 30 [28–32] | 0.46 |
| C-Reactive Protein (mg/dL)^ | 14.7 [3.5–21.4] n = 18 | 6.1 [1.7–11.2] n = 19 | 0.012 |
| Lactate dehydrogenase (U/L)^ | 306 [250–394] n = 19 | 351 [208–431] | 0.50 |

*Mean+SD

^Median [interquartile range]

¶statistically signifcant p-value<0.05

eGFR: estimated Glomerular Filtration Rate

**Table 3. Primary measure analyses.**

| \ | Case n = 20 | Control n = 20 | Odds Ratio (95% CI)† |
|---|---|---|---|
| Vitamin D deficient <20ng/ml, % Yes | 50% (10) | 15% (3) | 5.7 (1.2–26.4)* |
| Female | 50% (4) | 0% (0) | |
| Male | 50% (6) | 25% (3) | |
| APL antibody (any) Positive, % Yes | 90% (18) | 50% (10) | 7.4 (1.4–38.2)* |
| Female | 75% (6) | 62.5% (5) | |
| Male | 100% (12) | 41.7% (5) | |
| Lupus anticoagulant, % positive | 40% (8) | 5% (1) | 10.3 (1.3–79.5)* |
| Female | 62.5% (5) | 0% (0) | |
| Male | 25% (3) | 8.3% (1) | |
| Anticardiolipin IgM, % >12.5 | 80% (16) | 50% (10) | 3.4 (0.89–13.0) |
| Female | 50% (4) | 62.5% (5) | |
| Male | 100% (12) | 41.7% (5) | |
| Anticardiolipin IgG, % >15 | 25% (5) | 5% (1) | 5.0 (0.65–38.5) |
| Female | 0% (0) | 12.5% (1) | |
| Male | 41.7% (5) | 0% (0) | |

^APL = Antiphospholipid

* significant at $p < 0.05$

†Odds ratio and 95% CI determined by the Cochran Mantel–Haenszel test

Sub-analyses of APLa positive and negative patients categorized as vitamin D deficient or not were performed as illustrated in Fig 1. Patients who were both APLa positive and vitamin D deficient had significantly more clots compared to patients who were APLa positive with no vitamin D deficiency (100% vs 47.4% p = 0.01).

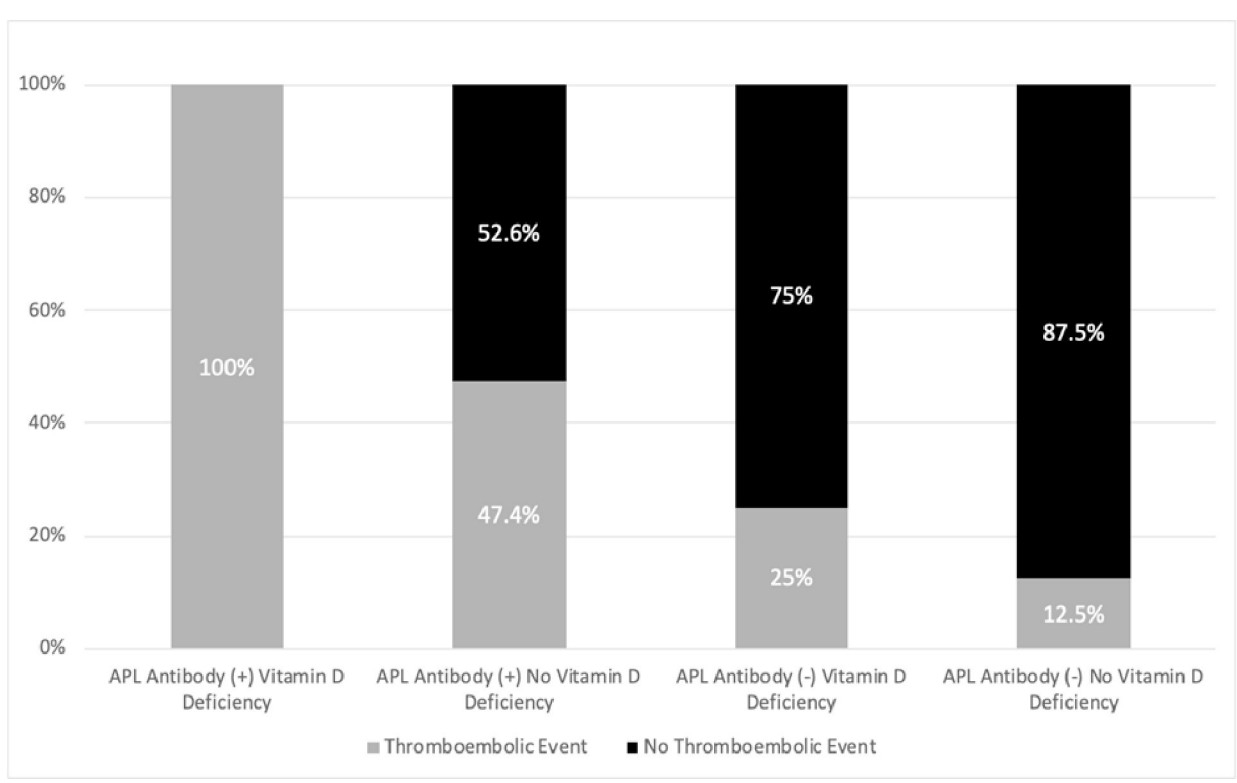

**Fig 1. Assessment of antiphospholipid antibodies and vitamin D combinations.**

Several studies have now reported the association of venous and/or arterial thromboembolic events with COVID-19. VTE incidence amongst COVID-19 patients in Wuhan, China admitted to the intensive care unit was reported to be 25% (20/81) [1]. Subsequently, a study on hospitalized COVID-19 patients in Italy showed that VTE events occurred in 21% of patients, with majority being PE, 2.5% had ischemic stroke and 1% had myocardial infarction [2]. A Dutch study then showed a 27% incidence of VTE, with majority (81%) being PE and 3.7% with arterial thrombotic events amongst COVID-19 patients admitted to intensive care unit [16]. Another study on the incidence of VTE in hospitalized COVID-19 patients showed cumulative incidences of VTE at day 7, 14, and 21 of 16% (95% CI:30–54), 33% (95% CI: 23–43), and 42% (95% CI:30–54) respectively, and in this study, VTE was significantly associated with mortality (hazard ratio (HR) 2.7; 95% CI:1.3–5.8) [17]. A meta-analysis of 102 studies (64, 503 patients) on the prevalence of venous and arterial thromboembolic events in patients with COVID-19 showed that the overall prevalence of VTE related to COVID-19 was 14.7% (95% CI: 12.1% - 17.6%), with the highest prevalence in patients admitted to the intensive care units (23.2% [95% CI: 17.5%-29.6%]) [18]. Based on these studies and our study results, the most common thrombotic events in hospitalized COVID-19 patients are VTE, especially PE.

Several mechanisms for the thrombotic manifestations in COVID-19 have been identified. COVID-19 is associated with pulmonary microvascular abnormalities such as intravascular fibrin deposition, perivascular monocyte infiltration, microthrombi formation and angiogenesis [19, 20]. Inflammation of pulmonary endothelial cells with disruption of membrane due to direct effects of the virus or due to host inflammatory effects are found in COVID-19 [21, 22]. Autopsy studies in COVID-19 have reported VTE in up to 50% of patients and with frequent development of DVT, PE in COVID-19 may be due to both immunothrombosis and conventional embolic mechanism [5, 19, 22]. Hypoxia has been proposed as a mechanism for endothelial dysfunction and hypercoagulability [23, 24]. Zuo et al., reported significantly elevated levels of calprotectin, a neutrophil activation marker, and neutrophil extracellular traps (NETs) formed by activated neutrophils in hospitalized COVID-19 patients with arterial or venous thrombosis compared to COVID-19 patients without thrombotic events [25]. NETs have been implicated in the pathogenesis of thrombosis in COVID-19 [26]. In our study, the neutrophil levels were significantly higher in the patients with thrombosis compared to patients without thrombosis. Cases were found to have statistically significant elevation of Alkaline Phosphatase as compared to controls which could be secondary to concomitant Vitamin D deficiency. A study by Bellastella et al., showed the correlation between Vitamin D and Alkaline Phosphatase due to possible involvement of Alkaline Phosphatase in the regulation of 25-Vitamin D-hydroxylase [27].

APLa has been connected to thrombosis, severe disease, and release of NETs in COVID-19 [8, 28, 29]. Monocyte and endothelial cell activation by APLa leads to overexpression of tissue factor and adhesion molecules causing thrombosis [15]. Anti-beta 2 glycoprotein 1 has been shown to upregulate both endothelial leucocyte adhesion molecule-1 (ELAM-1) and intracellular adhesion molecule-1 (ICAM-1) [30]. It has also been shown that APLa activates endothelial cells through toll-like receptors (TLRs), which induce inflammation [31]. A meta-analysis by Taha et al., reviewed 21 studies that included 1159 hospitalized patients with COVID-19 [9]. This study found that critically ill COVID-19 patients had higher levels of anticardiolipin antibodies than non-critically ill patients (28.8% vs 7.1%; p<0.0001), and similarly significantly higher levels of anti-beta-2 glycoprotein 1 antibodies (12.0% vs 5.8%, p<0.0001). The presence of APLa, however, was not associated with mortality, thrombosis, or mechanical ventilation.

Vitamin D has been shown to be a potent inhibitor of anti-beta-2 glycoprotein 1 antibody mediated tissue factor expression in endothelial cells and tumor necrosis factor or lipopolysaccharide induced tissue factor expression in monocytes [15, 32, 33]. In addition, vitamin D has

been shown to decrease the expression of adhesion molecules ELAM-1and ICAM-1, and TLR [34, 35]. The ability of vitamin D to reduce tissue factor and adhesion molecule expression and decrease inflammation by reducing expression of TLRs may explain the significantly lower rate of thrombosis in COVID-19 patients with APLa who have adequate vitamin D levels. The association of vitamin D deficiency and increased risk of VTE has also been shown in other conditions. Vitamin D deficiency was associated with development of DVT (OR 4.683, 95% CI 1.396–15.703) in patients with ischemic stroke [36]. High dose calcitriol was found to reduce venous and arterial thrombotic complications in a study of prostate cancer [37].

A retrospective cohort study found that vitamin D deficient status (25-hydroxycholecalciferol < 20 ng/mL or 1, 25-dihydroxycholecalciferol < 18 pg/mL) was associated with a higher risk of COVID-19 positivity (relative risk 1.77; 95% CI 1.12–1.81, P = 0.2) [38]. However, a study by Brandao et al found no correlation between Vitamin D levels and severity of COVID-19 disease [39]. Similarly, a study by Reis et al that included 220 patients with moderate to severe COVID-19 disease found no significant relationship between Vitamin D levels and the primary outcome of length of hospital stay and the secondary outcomes of mortality and mechanical ventilation [40]. A randomized placebo-controlled trial of short term administration of high dose (60,000 IU) cholecalciferol to asymptomatic or mildly symptomatic COVID-19 patients showed that a higher proportion of patients given high dose cholecalciferol became SARS-CoV-2 negative and fibrinogen levels were significantly lower in the patients given cholecalciferol [41]. In a randomized double-blind, placebo-controlled trial, a single high dose (200,000 IU) of vitamin $D_3$ given to hospitalized moderately to severely ill COVID-19 patients did not reduce the length of hospital stay, in-hospital mortality, admission to intensive care unit or mechanical ventilation need [42]. This study did not report incidence of thromboembolic events during hospitalization in patients given vitamin D supplementation and did not have APLa testing [42].

An open labelled, multi-center, randomized controlled trial on therapeutic versus prophylactic anticoagulation for hospitalized COVID-19 patients with elevated D-dimer, found that therapeutic anticoagulation did not improve outcomes but increased bleeding when compared to prophylactic anticoagulation [43]. A randomized clinical trial showed that intermediate-dose prophylactic anticoagulation when compared to standard dose prophylactic anticoagulation for COVID-19 patients admitted to the intensive care unit, did not lead to a significant difference in the primary efficacy outcome of a composite of venous or arterial thrombosis, extracorporeal membrane oxygenation need or 30 day-mortality [44]. Subsequently, investigators from REMAP-CAP, ACTIV-4a, ATTACC trials found that therapeutic anticoagulation with heparin in hospitalized COVID-19 patients improved the probability of survival to hospital discharge only in the non-critically ill but not in critically ill patients [45, 46]. Currently, National Institutes of Health (NIH) gives a weak recommendation for the use of therapeutic dose heparin in hospitalized COVID-19 patients who only require low flow oxygen and recommends against the use of therapeutic anticoagulation in critically ill COVID-19 patients in the intensive care units or requiring high flow oxygen [47].

In our study, 14 out of 15 PE, 2 out of 3 DVT, the ischemic stroke and NSTEMI were diagnosed on initial presentation to the hospital. 52% of patients with COVID-19 have been found to have at least one APLa [8]. We found that all our study patients with both APLa and vitamin D deficiency developed thrombotic complications. In patients with APLa without vitamin D deficiency, thrombotic complication was found only in 47.4% of patients. At the time of this writing, ClinicalTrials.gov lists 28 ongoing trials on the use of various anticoagulants in hospitalized COVID-19 (intensive care and floor) patients, outpatients with COVID-19 and COVID-19 patients post discharge. There were also 62 studies on the use of vitamin D in COVID-19 worldwide registered in ClinicalTrials.gov. Trials evaluating various

thromboprophylactic doses/medications across the spectrum of COVID-19 should consider screening for antiphospholipid antibodies and vitamin D deficiency and evaluate the effect of vitamin D supplementation and various thromboprophylactic regimens on the rates of COVID-19 related thromboembolism. Vitamin D sufficiency may have a role in preventing thromboembolic events in COVID 19, at least in patients with antiphospholipid antibodies, and this should be investigated further.

## Limitations

Our study has several limitations. Our sample size was small which affects the level of precision of study findings, as well as limits the ability to control for multiple confounding factors in the analyses. We did not follow up our patients after discharge from the hospital. We did not include any outpatients with mild or asymptomatic COVID-19. Race was not explored in this study. Our sample was predominantly White patients (70%) and vitamin D deficiency is more prevalent in African American patients [48]. However, we did account for the effect of sex variation related to our primary measure outcomes. We did not routinely screen all hospitalized COVID-19 patients with significantly elevated D-Dimer with venous duplex or CT angiogram for diagnosing asymptomatic DVT or small incidental PE. All our cases had symptomatic PE or DVT.

## Conclusions

In this small case control study, we found that hospitalized COVID-19 patients with thrombotic manifestations had significantly more coexistence of APLa positivity and vitamin D deficiency. Larger studies of the interaction of APLa and vitamin D in COVID-19 are warranted. In addition, rigorously conducted studies of vitamin D supplementation in COVID-19 of varied severity in both outpatient and in-patient setting, should assess the role of vitamin D in reducing the risk of thrombotic complications.

## Supporting information

**S1 Data.**
(XLSX)

## Acknowledgments

We thank Fabian Fregoli, MD, Chief Medical Officer, and Heidi Kromrei, PhD, Director of Medical Education of Saint Joseph Mercy Oakland for their staunch support of resident scholarly activity. We thank our IRB administrator Melody Dankha, and Research Nurses Jenny Romlein, BSN, RN, and Lynn Boomer, BSN, RN.

## Author Contributions

**Conceptualization:** Ruchi Shah, Anupam A. Sule, Geetha Krishnamoorthy.

**Data curation:** Ruchi Shah, Yaqub Nadeem Mohammed, Tracy J. Koehler, Jasmeet Kaur, Margarita Toufeili, Priyanjali Pulipati, Ahmed Alqaysi, Ali Khan, Mahrukh Khalid, Yi Lee, Parveen Dhillon, Anna Thao Dan, Nicholas Kumar, Anupam A. Sule, Geetha Krishnamoorthy.

**Formal analysis:** Ruchi Shah, Yaqub Nadeem Mohammed, Tracy J. Koehler, Priyanjali Pulipati, Ahmed Alqaysi, Ali Khan, Mahrukh Khalid, Yi Lee, Parveen Dhillon, Anna Thao Dan, Nicholas Kumar, Anupam A. Sule, Geetha Krishnamoorthy.

**Investigation:** Ruchi Shah, Yaqub Nadeem Mohammed, Jasmeet Kaur, Margarita Toufeili, Priyanjali Pulipati, Ahmed Alqaysi, Ali Khan, Mahrukh Khalid, Yi Lee, Anupam A. Sule, Geetha Krishnamoorthy.

**Methodology:** Ruchi Shah, Tracy J. Koehler, Jasmeet Kaur, Margarita Toufeili, Priyanjali Pulipati, Ahmed Alqaysi, Ali Khan, Mahrukh Khalid, Yi Lee, Anupam A. Sule, Geetha Krishnamoorthy.

**Project administration:** Ruchi Shah, Anupam A. Sule, Geetha Krishnamoorthy.

**Resources:** Yaqub Nadeem Mohammed, Jasmeet Kaur, Anupam A. Sule, Geetha Krishnamoorthy.

**Software:** Tracy J. Koehler.

**Supervision:** Ruchi Shah, Anupam A. Sule, Geetha Krishnamoorthy.

**Validation:** Ruchi Shah, Tracy J. Koehler, Anupam A. Sule, Geetha Krishnamoorthy.

**Visualization:** Ruchi Shah, Anupam A. Sule, Geetha Krishnamoorthy.

**Writing – original draft:** Ruchi Shah, Anupam A. Sule, Geetha Krishnamoorthy.

**Writing – review & editing:** Ruchi Shah, Yaqub Nadeem Mohammed, Tracy J. Koehler, Jasmeet Kaur, Margarita Toufeili, Priyanjali Pulipati, Ahmed Alqaysi, Ali Khan, Mahrukh Khalid, Yi Lee, Parveen Dhillon, Anna Thao Dan, Nicholas Kumar, Monica Bowen, Anupam A. Sule, Geetha Krishnamoorthy.

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
