## [Decision Letter · Decision Letter 0]

28 Mar 2022

PONE-D-21-28026Antiphospholipid antibodies and vitamin D deficiency in COVID-19 infection with and without venous or arterial thrombosis: A pilot case-control studyPLOS ONE

Dear Dr. Krishnamoorthy,

Thank you for submitting your manuscript to PLOS ONE. After careful consideration, we feel that it has merit but does not fully meet PLOS ONE’s publication criteria as it currently stands. Therefore, we invite you to submit a revised version of the manuscript that addresses the points raised during the review process.

Please resubmit the manuscript addressing the concerns raised by the reviewers.

We look forward to receiving your revised manuscript.

Kind regards,

Md Asiful Islam, Ph.D.

Academic Editor

PLOS ONE

Journal Requirements:

a) Did participants provide their written or verbal informed consent to participate in this study?

b) If consent was verbal, please explain i) why written consent was not obtained, ii) how you documented participant consent, and iii) whether the ethics committees/IRB approved this consent procedure

“Saint Joseph Mercy Oakland Hospital covered the cost of laboratory tests. Saint Joseph Mercy Hospital did not have any involvement in the study design; collection, analysis and interpretation of data; in the writing of the manuscript; and in the decision to submit for publication”

We note that you have provided additional information within the Funding Section that is not currently declared in your Funding Statement. Please note that funding information should not appear in the Funding section or other areas of your manuscript. We will only publish funding information present in the Funding Statement section of the online submission form.

Additional Editor Comments:

Please resubmit the manuscript addressing the concerns raised by the reviewers.

Reviewers' comments:

Reviewer's Responses to Questions

**Comments to the Author**

1. Is the manuscript technically sound, and do the data support the conclusions?

Reviewer #1: Yes

Reviewer #2: Yes

2. Has the statistical analysis been performed appropriately and rigorously? 

Reviewer #1: Yes

Reviewer #2: I Don't Know

3. Have the authors made all data underlying the findings in their manuscript fully available?

Reviewer #1: Yes

Reviewer #2: Yes

4. Is the manuscript presented in an intelligible fashion and written in standard English?

Reviewer #1: Yes

Reviewer #2: Yes

5. Review Comments to the Author

Reviewer #1: The manuscript of Shah et al. is well written and attempted to extrapolate some interesting findings. I have the following minor comments.

(i). Authors could revise the 1st paragraph of the introduction to include background details.

(ii). Organization of Table-2, and Table-3 data based on sex, could give a clear indication of sex-dependent variation as many biochemical parameters vary in males and females. Additionally, to compare the data variation with control, authors could use the start (*) sign, and mention the p-value in the table legend. Finally, in all the tables the abbreviated form should be listed in the table (1-4) legend.

(iii). Modification of the results, discussion and conclusion section based on the updated version of Table 2, and Table-3.

Reviewer #2: The study aimed to determine whether vitamin D deficiency is associated with thromboembolic events (TEs) both in the presence and absence of antiphospholipid antibodies (APLas). The paper identified that patients with both APLa positivity and vitamin D deficiency are significantly more prone to TEs than APLa positive patients with no vitamin D deficiency. The findings are significant because they can help set new parameters to characterize and treat COVID-19 patients with TEs. However, the paper is not suitable for publication in the present form and can only be accepted if the authors carefully make the following corrections and resubmit a revised version.

Major:

1. Page 8, line 6-12, Sample Size Calculation: The sample size calculation is irrational. To begin with, use the exact percentage from the previous literature where applicable - 52% and 91% as per your references. Reference 4 says that COVID-19 patients with prolonged aPTT had a 91% prevalence of “lupus anticoagulant” (LA) only. No information about other APLas is given in that article. Therefore, though LA is an APL antibody, the prevalence of LA cannot be interchangeably used with the prevalence of all APLas.

Moreover, Reference 3 says that 52% of “COVID-19 patients” have at least one APLa. There might be a considerable number of thrombotic patients among those patients, as a meta-analysis by Tan et al. (2021) found that 14.7% and 3.9% of COVID-19 patients had a venous thromboembolic event and an arterial thromboembolic event, respectively. Therefore, the patients of reference 3 are not comparable with the non-thrombotic patient group of your study. Devreese et al. (2020) can be a suitable reference for you, who found a 73% prevalence of APLas in non-thrombotic COVID-19 patients.

Furthermore, an elevated aPTT cannot be used synonymously with thrombosis. You may refer to Joncour et al. (2021) for a relevant comparison which found a 64% prevalence of APLas in thrombotic COVID-19 patients. You can see that the prevalence values of my referred studies are no way near to your calculated difference of 40%. In fact, a meta-analysis by Taha et al. (2021) clearly states, “There was no association between aPL positivity and disease outcomes including thrombosis.” Apart from these, no specific statistical formula was used to calculate the sample size. I recommend correcting the sample size calculation part with the help of an expert in statistics or completely removing the sample size calculation part from the manuscript if possible.

Minor:

1. Page 2, line 2, Conflict of interest: Correct the journal name.

2. Page 5, line 10-12: Zhang et al. is a clear outlier, as found in the meta-analysis by Taha et al. (2021). Also, a case series of three patients does not provide strong support behind any statement. I recommend removing the reference or replacing it with another one.

3. Page 5, line 17: reference 9 does not strongly support your statement. Consider changing or removing the reference.

4. Table 3: use ng/ml as vitamin D unit.

5. In table 4, add n=20 in both columns. Include a stacked column chart based on table 4 for a better visual representation.

6. In Discussion: Discuss why Alkaline phosphatase was found significantly higher in the case group than in control.

7. In Discussion: Discuss previous literature that oppose your claims too where applicable. For instance, mention article(s) that found no relationship between vitamin D deficiency and COVID-19 risk.

8. Refer preferentially to meta-analyses in cases of mentioning prevalence throughout your manuscript. E.g., thrombosis in COVID-19, vitamin D deficiency and COVID-19, etc.

9. Recheck the language of the manuscript and use English-checking software. Also, there are two places where a reference is placed after a full stop. Fix those.

6. PLOS authors have the option to publish the peer review history of their article (what does this mean?). If published, this will include your full peer review and any attached files.

Reviewer #1: No

Reviewer #2: No

---

## [Author Response · Author response to Decision Letter 0]

5 May 2022

Dear Dr.Islam, 

Thank you for inviting us to submit a revised draft of our manuscript entitled, "Antiphospholipid antibodies and vitamin D deficiency in COVID-19 infection with and without venous or arterial thrombosis: A pilot case-control study " to PLOS ONE. We also appreciate the time and effort you and the reviewers have dedicated to providing insightful feedback on ways to strengthen our paper. Thus, it is with great pleasure that we resubmit our article for further consideration. We have incorporated changes that reflect the detailed suggestions you have graciously provided. We also hope that our edits and the responses we provide below satisfactorily address all the issues and concerns you and the reviewers have noted.

To facilitate your review of our revisions, the following is a point-by-point response to the questions and comments delivered in your letter dated March 28th, 2022.

EDITOR SUGGESTIONS:

1. Formatting requirements:

Manuscript was changed in accordance with the PLOS ONE’s formatting guidelines.

We appreciate your input. A written informed consent was obtained from all the study participants. The study protocol was reviewed and approved by the Institutional Review Board Committee at Saint Joseph Mercy Oakland. Following sentence was added in the “Methods” section of the manuscript.

“Written informed consent was obtained from all the study participants in accordance with the consent procedure approved by the Institutional Review Board.”

3. Funding statement:

The below statement was removed from the revised manuscript. We instead request this to be added to the “Funding statement section” of the online submission form.

“Saint Joseph Mercy Oakland Hospital covered the cost of laboratory tests. Saint Joseph Mercy Hospital did not have any involvement in the study design; collection, analysis and interpretation of data; in the writing of the manuscript; and in the decision to submit for publication”.

4. Please include your full ethics statement in the ‘Methods’ section of your manuscript file. 

Please refer to #2. Full Ethics statement was added to the Methods section of revised manuscript as above.

REVIEWERS’ COMMENTS:

Reviewer #1: 

(i). Authors could revise the 1st paragraph of the introduction to include background details.

Answer: We have re-drafted the introduction section to establish a clearer focus. 

The initial section highlights few of the first studies demonstrating thrombosis in COVID-19, antiphospholipid antibodies in COVID-19 as well as Vitamin D in antiphospholipid antibody syndrome.

We added the following in the introduction:

“Zhang et al., first described a case series of three patients with COVID-19 with limb ischemia, who tested positive for anticardiolipin IgA and IgG and anti-beta-2 glycoprotein 1 IgA and IgG. Recently, more researchers found that in both antiphospholipid antibody syndrome and COVID-19 infection, systemic inflammatory response causes release of cytokines, and complement activation leading to endothelial damages and clot formation [4-6]. It is postulated that antiphospholipid antibodies (APLa) may be one of the mechanisms of thrombosis in patients with COVID-19 infection [7].”

“A meta-analysis that included 21 studies (total n=1159) reported that the most common APLa is lupus anticoagulant (50.7%); furthermore, 28.8% of critically ill patients had anticardiolipin (IgG or IgM) and 12% of them had anti-beta-2 glycoprotein 1 (IgG or IgM) [9].”

“In an animal study by Aihara et al., vitamin D receptors were shown to have a physiological role in the antithrombotic hemostasis [13]. A large prospective Danish study that included 18791 patients with a 30-year follow-up showed that reduced 25-hydroxy vitamin D concentrations are associated with an increased risk of venous thromboembolism [14].” 

(ii). Organization of Table-2, and Table-3 data based on sex, could give a clear indication of sex-dependent variation as many biochemical parameters vary in males and females. Additionally, to compare the data variation with control, authors could use the start (*) sign, and mention the p-value in the table legend. Finally, in all the tables the abbreviated form should be listed in the table (1-4) legend. 

Answer: We appreciate you bringing up such an interesting point to make our study better. We performed the analysis again for Table-3, which is our main primary measure analysis, using the Cochran-Mantel-Haenszel test to control for the effects of sex when deriving the odds ratio and confidence interval. We did find minor, but consistent, changes in the results after accounting for sex, and we updated the results as follows: 

“Accounting for the effects of sex, cases were 5.7 times more likely to be Vitamin D deficient (OR: 5.7, [95% CI: 1.2-26.4]) and 7.4 times more likely to have any one positive APLa (OR: 7.4 [ 95% CI: 1.4-38.2]; Table 3) and 10.3 times more likely to have a positive finding for lupus anticoagulant (OR: 10.3 [ 95% CI: 1.3-79.5]; Table 3). No other findings were significant between the groups”. 

Table 3 was updated to demonstrate the primary measure analyses while accounting for sex. 

With regards to Table 2, because of the small sample size which restricts the types of analyses that can be performed, it was decided to account for gender in the primary analyses only. This helps to control for gender differences in other lab values within our primary analyses as best we can, given the sample size. We are encouraged that our findings are consistent with the previous analyses performed. We also recognize this limitation and have acknowledged it within the study limitations section. Larger studies are needed to control for multiple confounders to explore the significant relationships further. 

Per your suggestion, we mentioned the p-value in the table legend. We also included abbreviations in all the table legends. 

(iii). Modification of the results, discussion and conclusion section based on the updated version of Table 2, and Table-3.

Answer: Yes, as discussed above, we made the modification and updated Table 3. Thank you for this suggestion. 

Reviewer #2: 

Major:

1. Page 8, line 6-12, Sample Size Calculation: The sample size calculation is irrational. To begin with, use the exact percentage from the previous literature where applicable - 52% and 91% as per your references. Reference 4 says that COVID-19 patients with prolonged aPTT had a 91% prevalence of “lupus anticoagulant” (LA) only. No information about other APLas is given in that article. Therefore, though LA is an APL antibody, the prevalence of LA cannot be interchangeably used with the prevalence of all APLas.

Moreover, Reference 3 says that 52% of “COVID-19 patients” have at least one APLa. There might be a considerable number of thrombotic patients among those patients, as a meta-analysis by Tan et al. (2021) found that 14.7% and 3.9% of COVID-19 patients had a venous thromboembolic event and an arterial thromboembolic event, respectively. Therefore, the patients of reference 3 are not comparable with the non-thrombotic patient group of your study. Devreese et al. (2020) can be a suitable reference for you, who found a 73% prevalence of APLas in non-thrombotic COVID-19 patients.

Furthermore, an elevated aPTT cannot be used synonymously with thrombosis. You may refer to Joncour et al. (2021) for a relevant comparison which found a 64% prevalence of APLas in thrombotic COVID-19 patients. You can see that the prevalence values of my referred studies are no way near to your calculated difference of 40%. In fact, a meta-analysis by Taha et al. (2021) clearly states, “There was no association between aPL positivity and disease outcomes including thrombosis.” Apart from these, no specific statistical formula was used to calculate the sample size. I recommend correcting the sample size calculation part with the help of an expert in statistics or completely removing the sample size calculation part from the manuscript if possible.

Answer: We really appreciate your input, and it is indeed an excellent suggestion. After extensive literature review and after reviewing the above-mentioned studies, we decided to remove the Sample size calculation from the manuscript. 

We rewrote that section with the following: 

“Ours is an exploratory study for hypothesis generation that concomitant presence of vitamin D deficiency and APLa in hospitalized patients with COVID-19 infection may increase the risk for venous and/or arterial thrombosis. Given the limited current literature on the role of concomitant APLa and vitamin D deficiency in hospitalized COVID-19 patients, we enrolled 20 patients in each arm (n=40) in total” 

Minor:

1. Page 2, line 2, Conflict of interest: Correct the journal name.

Answer: We are extremely sorry for the mistake. Journal name was corrected in the revised manuscript.

2. Page 5, line 10-12: Zhang et al. is a clear outlier, as found in the meta-analysis by Taha et al. (2021). Also, a case series of three patients does not provide strong support behind any statement. I recommend removing the reference or replacing it with another one.

Answer: We agree with your suggestion that a case series of three patients does not provide strong support, however, it was one of the first case series highlighting possible correlation between COVID-19 and Antiphospholipid antibodies published in NEJM. Hence, we mentioned it in the introduction paragraph, and then replaced it with other larger studies/meta-analysis in the discussion.

Please find below the updated statement as part of introduction:

“Zhang et al., first described a case series of three patients with COVID-19 with limb ischemia, who tested positive for anticardiolipin IgA and IgG and anti-beta-2-glycoprotein 1 IgA and IgG.”

We included the study by Taha et al., as suggested by you in the discussion as follows:

A meta-analysis by Taha et al., reviewed 21 studies that included 1159 hospitalized patients with COVID-19 [9]. This study found that critically ill COVID-19 patients had higher levels of anticardiolipin antibodies than non-critically ill patients (28.8% vs 7.1%; p<0.0001), and similarly significantly higher levels of anti-beta-2 glycoprotein 1 antibodies (12.0% vs 5.8%, p<0.0001). The presence of APLa , however, was not associated with mortality, thrombosis, or mechanical ventilation. 

3. Page 5, line 17: reference 9 does not strongly support your statement. Consider changing or removing the reference.

Answer: We agree with you. We removed the above reference (9) from our manuscript. 

4. Table 3: use ng/ml as Vitamin D unit.

Answer: Thank you for the suggestion, we updated the unit in Table 3 of the revised manuscript.

5. In table 4, add n=20 in both columns. Include a stacked column chart based on table 4 for a better visual representation.

Answer: Thank you for providing these insights. Adding a visual presentation definitely helps to explain our results. We included the data represented in table 4 of the original manuscript as a stacked column chart in the revised manuscript. 

6. In Discussion: Discuss why Alkaline phosphatase was found significantly higher in the case group than in control.

Answer: That is an interesting query. During our literature review, we were able to find correlation between Vitamin D deficiency and Alkaline Phosphatase. Our cases did have statistically significant Vitamin D deficiency as compared to our controls which would further explain the elevated Alkaline phosphatase levels amongst cases. We included the same in the revised manuscript as below:

“Cases were found to have statistically significant elevation of Alkaline Phosphatase as compared to controls which could be secondary to concomitant Vitamin D deficiency. A study by Bellastella et al., showed the correlation between Vitamin D and Alkaline Phosphatase due to possible involvement of Alkaline Phosphatase in the regulation of 25-Vitamin D-hydroxylase.”

7. In Discussion: Discuss previous literature that oppose your claims too where applicable. For instance, mention article(s) that found no relationship between vitamin D deficiency and COVID-19 risk.

Answer: You have raised an important point. While we were able to find studies supporting correlation between Vitamin D deficiency and COVID-19 risk/severity, we also found studies which did not. We included the same in our revised manuscript as below:

“However, a study by Brandao et al found no correlation between vitamin D levels and severity of COVID-19 disease. Similarly, the Reis et al study included 220 patients with moderate to severe COVID-19 disease found no significant relationship between vitamin D levels and the primary outcome of length of hospital stay and the secondary outcomes of mortality and mechanical ventilation. “

8. Refer preferentially to meta-analyses in cases of mentioning prevalence throughout your manuscript. E.g., thrombosis in COVID-19, vitamin D deficiency and COVID-19, etc.

Answer: We re-structured the introduction and discussion part of our manuscript to include meta-analyses. We included the study by Taha et al., as suggested by you and we included a large meta-analysis on the prevalence of VTE in patients with COVID-19 by Tan et al. as shown below:

A meta-analysis by Taha et al., reviewed 21 studies that included 1159 hospitalized patients with COVID-19 [9]. This study found that critically ill COVID-19 patients had higher levels of anticardiolipin antibodies than non-critically ill patients (28.8% vs 7.1%; p<0.0001), and similarly significantly higher levels of anti-beta-2 glycoprotein 1 antibodies (12.0% vs 5.8%, p<0.0001). The presence of APLa, however, was not associated with mortality, thrombosis, or mechanical ventilation. 

A meta-analysis of 102 studies (64, 503 patients) on the prevalence of venous and arterial thromboembolic events in patients with COVID-19 showed that the overall prevalence of VTE related to COVID-19 was 14.7% (95% CI: 12.1% - 17.6%), with the highest prevalence in patients admitted to the intensive care units (23.2% [95% CI: 17.5%-29.6%]).

9. Recheck the language of the manuscript and use English-checking software. Also, there are two places where a reference is placed after a full stop. Fix those.

Answer: Yes, we rechecked using Grammarly for checking grammar. 

Other changes:

Due to more evidence being available and changes being made to the degree of anticoagulation for COVID-19 patients, we updated the evidence and updated the manuscript reflecting the current NIH recommendation on the degree of anticoagulation as below:

“Subsequently, investigators from REMAP-CAP, ACTIV-4a, ATTACC trials found that therapeutic anticoagulation with heparin in hospitalized COVID-19 patients improved the probability of survival to hospital discharge only in the non-critically ill but not in critically ill patients [45, 46]. Currently, National Institutes of Health (NIH) gives a weak recommendation for the use of therapeutic dose heparin in hospitalized COVID-19 patients who only require low flow oxygen and recommend against the use of therapeutic anticoagulation in critically ill COVID-19 patients in ICU or requiring high flow oxygen [47].”

Additionally, the number of clinical trials evaluating various anticoagulation protocols and vitamin D supplementation in COVID-19 were updated as follows:

“At the time of this writing, ClinicalTrials.gov lists 28 ongoing trials on the use of various anticoagulants in hospitalized COVID-19 (intensive care and floor) patients, outpatients with COVID-19 and COVID-19 patients post discharge. There were also 62 studies on the use of vitamin D in COVID-19 worldwide registered in ClinicalTrials.gov”

CONCLUDING REMARKS: Again, thank you very much for giving us the opportunity to strengthen our manuscript with your valuable comments and queries. We have worked hard to incorporate your feedback and hope that these revisions persuade you to accept our submission.

Sincerely,

Geetha Krishnamoorthy

Corresponding Author

Department of Internal Medicine, Saint Joseph Mercy Oakland

Email: Geetha.krishnamoorthy@stjoeshealth.org

Ruchi Shah

Department of Internal Medicine, Saint Joseph Mercy Oakland

Email: ruchi.shah001@stjoeshealth.org

---

## [Decision Letter · Decision Letter 1]

23 May 2022

Antiphospholipid antibodies and vitamin D deficiency in COVID-19 infection with and without venous or arterial thrombosis: A pilot case-control study

PONE-D-21-28026R1

Dear Dr. Krishnamoorthy,

We’re pleased to inform you that your manuscript has been judged scientifically suitable for publication and will be formally accepted for publication once it meets all outstanding technical requirements.

Kind regards,

Md Asiful Islam, Ph.D.

Academic Editor

PLOS ONE

Additional Editor Comments (optional):

Reviewers' comments:

Reviewer's Responses to Questions

**Comments to the Author**

1. If the authors have adequately addressed your comments raised in a previous round of review and you feel that this manuscript is now acceptable for publication, you may indicate that here to bypass the “Comments to the Author” section, enter your conflict of interest statement in the “Confidential to Editor” section, and submit your "Accept" recommendation.

Reviewer #1: All comments have been addressed

Reviewer #2: All comments have been addressed

2. Is the manuscript technically sound, and do the data support the conclusions?

Reviewer #1: Yes

Reviewer #2: Yes

3. Has the statistical analysis been performed appropriately and rigorously? 

Reviewer #1: Yes

Reviewer #2: Yes

4. Have the authors made all data underlying the findings in their manuscript fully available?

Reviewer #1: Yes

Reviewer #2: Yes

5. Is the manuscript presented in an intelligible fashion and written in standard English?

Reviewer #1: Yes

Reviewer #2: Yes

6. Review Comments to the Author

Reviewer #1: I am happy with the current version of this manuscript. One minor comment, in the introduction part author should combine paragraph 1 and 2, and paragraph 3 and 4 together.

Reviewer #2: The authors have addressed all my comments properly in their revised version of the manuscript. Therefore, I recommend accepting the current version of the manuscript for publication.

7. PLOS authors have the option to publish the peer review history of their article (what does this mean?). If published, this will include your full peer review and any attached files.

Reviewer #1: No

Reviewer #2: No

---

## [Editor Report · Acceptance letter]

5 Jul 2022

PONE-D-21-28026R1 

Antiphospholipid antibodies and vitamin D deficiency in COVID-19 infection with and without venous or arterial thrombosis: A pilot case-control study 

Dear Dr. Krishnamoorthy:

I'm pleased to inform you that your manuscript has been deemed suitable for publication in PLOS ONE. Congratulations! Your manuscript is now with our production department. 

Kind regards, 

on behalf of

Dr. Md Asiful Islam 

Academic Editor

PLOS ONE